# Cell-Specific Effects of Insulin in a Murine Model of Restenosis Under Insulin-Sensitive and Insulin-Resistant Conditions

**DOI:** 10.3390/cells13161387

**Published:** 2024-08-20

**Authors:** Marel Gonzalez Medina, Zhiwei Liu, Johny Wang, Cindy Zhang, Sarah B. Cash, Carolyn L. Cummins, Adria Giacca

**Affiliations:** 1Department of Physiology, Temerty Faculty of Medicine, University of Toronto, Toronto, ON M5S 1A8, Canada; mag658@miami.edu (M.G.M.); zhiweispencer.liu@mail.utoronto.ca (Z.L.); johny.wang@mail.utoronto.ca (J.W.); cindyxinyue.zhang@mail.utoronto.ca (C.Z.); 2Department of Pharmaceutical Sciences, Leslie Dan Faculty of Pharmacy, University of Toronto, Toronto, ON M5S 3M2, Canada; sarah.cash@mail.utoronto.ca (S.B.C.); carolyn.cummins@utoronto.ca (C.L.C.); 3Department of Medicine, Temerty Faculty of Medicine, University of Toronto, Toronto, ON M5S 3H2, Canada; 4Institute of Medical Science, University of Toronto, Toronto, ON M5S 3H2, Canada; 5Banting and Best Diabetes Centre, University of Toronto, Toronto, ON M5G 2C4, Canada

**Keywords:** insulin, insulin resistance, mouse, neointimal growth, vascular endothelial cell, vascular smooth muscle cell

## Abstract

Restenosis following percutaneous revascularization is a major challenge in patients with insulin resistance and diabetes. Currently, the vascular effects of insulin are not fully understood. In vitro, insulin’s effects on endothelial cells (ECs) are beneficial, whereas on vascular smooth muscle cells (SMCs), they are mitogenic. We previously demonstrated a suppressive effect of insulin on neointimal growth under insulin-sensitive conditions that was abolished in insulin-resistant conditions. Here, we aimed to determine the cell-specific effects of insulin on neointimal growth in a model of restenosis under insulin-sensitive and insulin-resistant conditions. Vascular cell-specific insulin receptor (IR)-deficient mice were fed a low-fat diet (LFD) or a high-fat, high-sucrose diet (HFSD) and implanted with an insulin pellet or vehicle prior to femoral artery wire injury. In insulin-sensitive conditions, insulin decreased neointimal growth only in controls. However, under insulin-resistant conditions, insulin had no effect in either control, EC-specific or SMC-specific IR-deficient mice. These data demonstrate that EC and SMC IRs are required for the anti-restenotic effect of insulin in insulin-sensitive conditions and that, in insulin resistance, insulin has no adverse effect on vascular SMCs in vivo.

## 1. Introduction

Cardiovascular disease (CVD) resulting from atherosclerosis remains the leading cause of mortality globally [1]. Percutaneous transluminal angioplasty and stenting have been effective in the treatment of atherosclerotic vessels [2]; however, these procedures may fail due to postangioplasty restenosis [2]. Although the rate of in-stent restenosis following percutaneous revascularization has decreased to less than 10% in past decades due to the development of drug-eluting stents (DESs) [3], this success rate has not been achieved in patients with insulin resistance and type 2 diabetes [4,5].

Insulin has been shown to directly exert multiple actions in the cardiovascular system in addition to its metabolic actions [6,7]. Notably, it is still debated whether insulin is protective or detrimental in the vasculature [6,7]. Early in vitro studies demonstrated that supraphysiological concentrations of insulin promoted vascular smooth muscle cell (SMC) migration and proliferation mediated via the mitogen-activated protein kinase (MAPK) pathway, suggesting insulin’s adverse effects [8]. However, in vivo studies have shown that the effects of insulin are mostly beneficial (i.e., vasodilatory, anti-inflammatory, anti-atherogenic) [9,10] and mediated via the phosphatidylinositol 3-kinase (PI3K)/protein kinase B (Akt)/endothelial nitric oxide synthase (eNOS) pathway [11]. Indeed, our previous studies in rodent models of restenosis showed that administration of insulin was effective in reducing neointimal hyperplasia, the main factor for in-stent restenosis [12,13]. Further, we have recently found that in insulin-sensitive, chow-fed mice, deletion of either the endothelial or the SMC insulin receptor (IR) abolishes the effect of insulin to decrease neointimal growth [14].

Importantly, the cell-specific effects of insulin may be different in insulin-resistant conditions. During insulin resistance, it has been hypothesized that the vasculoprotective PI3K/Akt/eNOS pathway is impaired in endothelial cells (ECs), whereas the mitogenic MAPK pathway remains intact in vascular SMCs (‘selective insulin resistance’) [15]. A previous study by our group found that the insulin-induced decrease in neointimal growth was abolished in high-fat-fed [13] or sucrose-fed [16] rats. Thus, it is possible that in insulin-resistant conditions, the beneficial effect of insulin in ECs is decreased and counterbalanced by an adverse effect in vascular SMCs. These dual effects could be unmasked in conditions of cell-specific IR deficiency. For instance, when a high-fat diet was used to induce insulin resistance, other authors have reported that IR deletion in SMCs decreases the neointimal area, suggesting a growth-promoting effect of insulin in vascular SMCs [17]. However, this study yielded significant differences in the neointimal area but not in the intima to media (I/M) ratio and failed to examine the effect of insulin treatment in this model. Therefore, further investigation of the cell-specific effects of insulin in murine models of restenosis is required.

In the present study, we determined the cell-specific effects of insulin on neointimal growth in an in vivo murine model of restenosis under insulin-sensitive and insulin-resistant conditions by using the CreERT2-loxP-mediated recombination system, which can selectively and conditionally induce IR deletion in ECs or SMCs without affecting tissue development.

## 2. Materials and Methods

### 2.1. Animal Model and Animal Experiments

All procedures were in accordance with the Canadian Council of Animal Care Standards and were approved by the Animal Care Committee of the University of Toronto. Mice were housed in the University of Toronto’s Department of Comparative Medicine and had access to ad libitum food and water.

Mice carrying an IR gene, in which exon 4 is flanked by loxP sites, i.e., IR floxed mice, were obtained from Jackson Labs (stock #006955). Mice with transgene CreERT2 under the control of Cdh5 (vascular endothelial cadherin) [18] or SMMHC (smooth muscle heavy chain) [19] promoters, i.e., Cre-positive mice, were obtained from Taconic (stock #13073-M) and provided by Dr. Stefan Offermanns, respectively. IR floxed mice were crossed with heterozygous Cdh5-Cre- or SMMHC-Cre-positive mice (IR^f/f^-Cre(+) mice) to obtain IR^f/f^-Cdh5-Cre(+) mice or IR^f/f^-SMMHC-Cre(+) mice, respectively. Littermate Cre-negative mice also carrying a homozygous floxed IR gene (IR^f/f^-Cre(−) mice) were used as Cre-negative controls. In a subset of experiments, heterozygous Cdh5-Cre or SMMHC-Cre-positive mice with a wild-type IR gene (IR^w/w^-Cre(+) mice) were used as Cre-positive controls. Male mice were used for this study because the SMMHC-Cre transgene used is located on the Y chromosome [20]. All mouse strains were on a C57BL/6N genetic background. The experimental protocol is described in Figure 1.

### 2.2. Diet Treatment

Mice were fed with a high-saturated-fat and high-sucrose diet (HFSD) (45% kcal fat, 50% of carbohydrate content as sucrose; Research Diets, New Brunswick, NJ, USA, Catalog #12451) or a low-fat diet (LFD) (10% kcal fat, no sucrose, same protein and micronutrient composition as the HFSD; Research Diets, New Brunswick, NJ, USA, Catalog #12450K) at four weeks of age. The HFSD mimics the high-fat, high-refined-sugar Western diet in humans and induces insulin resistance and glucose intolerance in C57BL/6 mice [21].

### 2.3. Generation of Cell-Specific IR-Deficient Mice 

Tamoxifen (Sigma-Aldrich, St. Louis, MO, USA; 50 mg/kg body weight) dissolved in corn oil was injected i.p. once every 24 h for 10 days [17] to delete the IR from ECs in IR^f/f^-Cdh5-Cre(+) mice or SMCs in IR^f/f^-SMMHC-Cre(+) mice. Cre-negative (IR^f/f^-Cre (−)) and Cre-positive (IR^w/w^-Cre (+)) control mice also received i.p. tamoxifen injection once every 24 h for 10 days. All mice were injected at five weeks of age.

### 2.4. Insulin Pellet Implantation and Surgical Procedure

Insulin-releasing pellets (LinBit 0.05 U/day; recombinant synthetic human insulin, micro-recrystallized palmitic acid; 13 ± 2 mg/pellet, LinShin Canada Inc., Toronto, ON, Canada) or blank pellets (100% micro-recrystallized palmitic acid) were implanted subcutaneously in mice under isoflurane gas at eight weeks of age. Our goal was to achieve similar plasma insulin elevation (i.e., ~4-fold basal) to that observed in our previous studies of the effect of insulin on atherosclerosis [22]. Drinking water was replaced with 20% glucose water throughout the experimental period to maintain normoglycemia in insulin-treated mice, a treatment we previously showed not to affect intimal hyperplasia [16].

Three days following the subcutaneous insulin pellet implantation procedure, mice were anesthetized with isoflurane and underwent femoral artery wire injury using a dissecting microscope (SMZ-800, Nikon, Tokyo, Japan). Briefly, the deep femoral artery was exposed and ligated distally, followed by a transverse arteriotomy distal to the bifurcation to allow insertion of a straight-spring guide wire (0.38 mm in diameter, C-SF-15-15; Cook Medical, Bloomington, IN, USA) into the common femoral artery as previously described [23]. The wire was left in the femoral artery for 1 min to denude and dilate the artery. Then, the wire was removed and the proximal portion of the deep femoral artery was ligated.

### 2.5. Blood Glucose Assessment and Blood Sample Collection

Fasting and random blood glucose were measured via the tail vein once a week after femoral artery wire injury using a glucometer (OneTouch UltraMini Blood Glucose Meter, LifeScan Canada Ltd, Quebec, QC, Canada.) and values were converted to plasma glucose assuming a normal hematocrit. For the measurement of fasting glucose levels, mice had no access to food for a duration of four hours. Insulin-treated mice were allowed 20% glucose water for the initial two hours of fasting to avoid hypoglycemia. In addition, blood samples were collected via cardiac puncture at the time of sacrifice for analysis of metabolic parameters, such as plasma insulin (ELISA obtained from ALPCO, Salem, NH, USA, Catalog #80-INSMS-E01, E10), insulin-like growth factor 1 (IGF-1) (ELISA obtained from R&D systems, Minneapolis, MN, USA, Catalog #MG100), vascular endothelial growth factor (VEGF) (ELISA obtained from R&D systems, Minneapolis, MN, USA, Catalog #MMV00-1), triglyceride (TG), free fatty acids (FFAs) (both colorimetric assay technique, Wako Diagnostics, Osaka, Japan), and total cholesterol (Infinity Cholesterol Kit obtained from Thermo Scientific, Waltham, MA, USA, Catalog #TR13421). 

### 2.6. Insulin Tolerance Test

Insulin tolerance tests were conducted in floxed control and cell-specific IR-deficient mice five days prior to sacrifice to examine the effects of diet and insulin treatment and the potential systemic effects of EC- or SMC-specific IR deficiency. After 4 h, fasting glucose levels were taken and a bolus of insulin (Humulin R, 0.5 U/kg body weight) was injected i.p. Plasma glucose levels were measured at 15, 30, 60, and 120 min post-injection, and the area under the curve (AUC) of the measurements was calculated. In addition, plasma glucose as a percentage of basal glucose was plotted over time for each group and the AUC of the measurements was calculated.

### 2.7. Fixed Vessel Collection and Morphological Analysis

Injured and uninjured femoral arteries were collected following perfusion fixation with PBS and 10% neutral formalin 28 days following femoral artery wire injury. Six serial cross-sections of the paraffin-embedded injured femoral arteries at 0.1 mm intervals were stained with Elastin van Gieson (EVG) and the average was used as a single data point. Uninjured or thrombosed sections were excluded. The cross-sections were digitized using a high-resolution scanning instrument (Hamamatsu Nanozoomer, Shizuoka, Japan) and analyzed using Image J software (Version 2.14.0, National Institutes of Health, Bethesda, MD, USA). 

The neointimal area was defined as the area encompassed between the internal elastic lamina and the lumen. The medial area was measured as the area encompassed by the external elastic lamina subtracted by the area enclosed by the internal elastic lamina. The ratio of intima to media (I/M ratio), which takes into consideration vessel size, was calculated by dividing the neointimal area by the medial area.

### 2.8. Immunofluorescence

Cross-sections of uninjured femoral arteries were used for immunofluorescence analysis. Tissue sections 4 µm thick were stained sequentially with Anti-IRβ, CD31, and αSMA. Antigen retrieval was performed using TRIS-EDTA (pH 9.0) for 7 min in a pressure cooker followed by a cool-down period of 10 min at room temperature. After blocking steps, the sections were incubated with the primary antibodies under the indicated conditions and stained with their respective secondary antibodies for 1 h at room temperature. The following primary antibodies were used: anti-IRβ antibody (Santa Cruz, Dallas, TX, USA, Catalog #sc-711, RRID: AB_631835, 1:100 dilution, 2 h incubation at room temperature, visualized with Goat anti-Rabbit IgG  +  Alexa Fluor 555, Catalog #A21428, 1:200 dilution, from Invitrogen, Waltham, MA, USA); Rabbit anti-CD31 antibody (Abcam, Cambridge, UK, Catalog #ab28364, RRID: AB_726362, 1:100 dilution, overnight incubation at 4 °C, visualized using Goat anti-Rabbit IgG + Alexa Fluor 647, Catalog #A21247, 1:200 dilution, from Invitrogen, Waltham, MA, USA); and, finally, anti-α-SMA (Smooth Muscle Actin) antibody, which is FITC conjugated (Abcam, Cambridge, UK, Catalog #ab202295, RRID: AB_ 2890884, 1:300 dilution, 90-min incubation at room temperature). Nuclei were stained with DAPI (Sigma-Aldrich, St. Louis, MO, USA, Catalog #D9542, 100 ng/mL). Stained cross-sections were digitized using the Olympus VS-120 slide scanner (Olympus, Shinjuku, Tokyo, Japan) and images were analyzed using OlyVIA software (Version 2.9.1, Olympus, Tokyo, Japan) and Image J software (Version 2.14.0, National Institutes of Health, Bethesda, MD, USA).

### 2.9. Statistical Analysis

Statistical calculations were conducted using GraphPad Prism 9 Software. Data were represented as mean ± standard error of the mean (SEM). An assessment of normality was performed by using Kolmogorov–Smirnov and Shapiro–Wilk tests. Accordingly, the unpaired t-test (parametric test for normal distribution) or Mann–Whitney test (non-parametric test for non-normal distribution) were used for comparisons of two groups. Differences between multiple groups were analyzed by one-way, two-way (genotype, diet), or three-way (genotype, diet, treatment) ANOVA followed by Tukey’s post hoc test for normal distribution or by a Kruskal–Wallis test followed by Dunn’s post hoc test for non-normal distribution. Statistical significance was accepted at *p* < 0.05. Power analysis showed that using our coefficient of variation of 20% and the *n* studied, we had a power equal to or greater than 0.8 to detect a 50% difference in the I/M ratio.

## 3. Results

### 3.1. Generation of Cell-Specific IR-Deficient Mice

Firstly, we conducted immunofluorescence analysis to assess IR expression in ECs and SMCs of the uninjured femoral arteries in tamoxifen-treated IR^f/f^-Cdh5-Cre(+) and IR^f/f^-Cdh5-Cre(−) mice. As observed in Figure 2A–D, IR^f/f^-Cdh5-Cre(+) mice had a reduction of more than 90% in IR expression in ECs, but not in SMCs, relative to IR^f/f^-Cdh5-Cre(−) mice.

We also assessed IR expression in ECs and SMCs of the uninjured femoral arteries in tamoxifen-treated IR^f/f^-SMMHC-Cre(+) and IR^f/f^-SMMHC-Cre(−) mice. Immunofluorescence analysis revealed that IR^f/f^-SMMHC-Cre(+) mice had a reduction of more than 80% in IR expression in SMCs, but not in the ECs, compared to IR^f/f^-SMMHC-Cre(−) mice (Figure 2E–H). The HFSD did not affect IR expression in ECs or SMCs (Appendix A).

### 3.2. Metabolic Parameters

Body weight and fasted and fed plasma glucose levels were significantly higher in HFSD-fed mice regardless of genotype or insulin treatment (Table 1 and Table 2).

Insulin treatment increased plasma insulin concentrations by approximately four-fold regardless of diet or genotype. Notably, insulin levels were not increased in HFSD-fed vehicle-treated mice compared to LFD-fed mice despite HFSD-induced hyperglycemia and insulin resistance (see Section 3.4), suggesting that HFSD-fed mice had impaired β-cell function. Further, the HFSD increased TG, FFA, total cholesterol, and IGF-1 levels in control and EC- or SMC-specific IR-deficient mice, while insulin treatment had a decreasing effect on TG and FFA. No differences were observed in the VEGF levels, regardless of genotype, treatment, or diet.

### 3.3. Cell-Specific Effects of Insulin on Neointimal Growth under Insulin-Sensitive Conditions

#### 3.3.1. Insulin’s Anti-Restenotic Effect Is Abolished in LFD-Fed EC-Specific IR-Deficient Mice

The morphological analysis of injured femoral arteries indicated that in the absence of insulin treatment (i.e., after vehicle treatment), the neointimal area and I/M ratio were comparable between tamoxifen-treated IR^f/f^-Cdh5-Cre(−) control mice and IR^f/f^-Cdh5-Cre(+)EC-specific IR-deficient mice (Figure 3A–C). Further, in IR^f/f^-Cdh5-Cre(−) control mice, insulin treatment led to a decrease in the neointimal area (Figure 3B) and I/M ratio (Figure 3C). Conversely, in IR^f/f^-Cdh5-Cre(+)EC-specific IR-deficient mice, the neointimal area and I/M ratio were not significantly affected by insulin treatment (Figure 3A–C).

#### 3.3.2. Insulin’s Anti-Restenotic Effect Is Abolished in LFD-Fed SMC-Specific IR-Deficient Mice

In the absence of insulin treatment (i.e., after vehicle treatment), the neointimal area and I/M ratio were similar between tamoxifen-treated IR^f/f^-SMMHC-Cre(−) control mice and IR^f/f^-SMMHC-Cre(+) SMC-specific IR-deficient mice (Figure 4A–C). While in IR^f/f^-SMMHC-Cre(−) control mice, insulin treatment decreased the neointimal area and I/M ratio, insulin failed to have the same effect in IR^f/f^ SMMHC-Cre(+) SMC-specific IR-deficient mice (Figure 4A–C).

#### 3.3.3. Insulin Treatment Reduces Neointimal Growth in LFD-Fed IR^W/W^-Cre(+) Control Mice

To rule out that Cre expression itself interferes with the effect of insulin on neointimal growth, we studied tamoxifen-treated Cre-positive wild-type mice (IR^w/w^-Cdh5-Cre(+) and IR^w/w^-SMMHC-Cre (+)), which are other control groups. We found that insulin treatment led to a decrease in the neointimal area (Appendix A) and I/M ratio (Appendix A) in IR^w/w^-Cdh5-Cre(+) control mice compared to vehicle treatment (Appendix A). Similarly, insulin treatment led to a decrease in neointimal area and I/M ratio in IR^w/w^-SMMHC-Cre(+) control mice, compared to vehicle treatment (Appendix A).

### 3.4. HFSD Induces Insulin Resistance in Control and Cell-Specific IR-Deficient Mice

To examine the effects of diet, insulin treatment, and the potential systemic effects of EC- or SMC-specific IR deficiency, an ITT was conducted five days prior to sacrifice. Since no differences related to genotype were observed (Appendix A) in accordance with our previous study [14], IR-deficient and control mice were grouped together. Plasma glucose levels following the intraperitoneal injection of insulin at a dose of 0.5 U/kg were only different between LFD-fed and HFSD-fed mice irrespective of prior insulin treatment, indicating that the HFSD induces systemic insulin resistance in EC- or SMC-specific IR-deficient and control mice and that prior insulin treatment had no effect (Figure 5A–D and Figure 6A–D).

### 3.5. Cell-Specific Effects of Insulin on Neointimal Growth under Insulin-Resistant Conditions

In HFSD-fed mice, we found no significant difference in the neointimal area and I/M ratio between the vehicle-treated and insulin-treated tamoxifen-injected IR^f/f^-Cdh5-Cre(−) control mice (Figure 7A–C). Similarly, these parameters were comparable between the vehicle and insulin groups in the tamoxifen-injected IR^f/f^-Cdh5-Cre(+)EC-specific IR-deficient mice, which overall indicates that in insulin-resistant conditions, the beneficial effect of insulin on neointimal growth is abolished but not reversed, even when the endothelial-dependent beneficial effects of insulin are abolished. Additionally, these parameters were comparable between the vehicle and insulin groups in the tamoxifen-injected IR^f/f^-SMMHC-Cre(+) SMC-specific IR-deficient mice (Figure 8A–C), which overall indicates that in insulin-resistant conditions, the lack of effect of insulin on vascular SMCs does not attenuate neointimal growth, i.e., insulin does not have an adverse effect on vascular SMCs.

The morphological analysis of injured femoral arteries in HFSD-fed tamoxifen-injected IR^w/w^-Cdh5-Cre(+) (Appendix A) and IR^w/w^-SMMHC-Cre(+) control mice (Appendix A) showed no significant difference in the neointimal area and I/M ratio between vehicle and insulin-treated groups, in accordance with the results in IR^f/f^ Cre(−) mice.

Although the intimal area and I/M ratio appeared to be similar in LFD- and HFSD-fed vehicle-treated mice (compare Figure 3 and Figure 4 with Figure 7 and Figure 8), there was a significant difference in the neointimal area (greater with HFSD, *p* < 0.01) and a trend toward a difference in I/M ratio (*p* = 0.11) following a two-way ANOVA (diet and genotype).

## 4. Discussion

Our current study on LFD-fed mice confirmed that both EC- and SMC-specific IRs contribute to the effect of insulin in decreasing neointimal growth in insulin-sensitive conditions. These results are consistent with our previous findings in chow-fed mice [14], where 0.1 U insulin/day was used, elevating insulin levels more than six-fold. Interestingly, we herein found that in LFD-fed mice, a lower and thus more physiological dose of insulin (0.05 U/day) elevating insulin levels by approximately four-fold recapitulated the beneficial effects of 0.1 U insulin/day to decrease neointimal growth after arterial injury. This elevation in insulin levels is similar to that observed in our previous study on atherosclerosis where the same insulin dose was used (0.05 U/day) [22]. Since both insulin doses (0.05 and 0.1 U/day) resulted in similar reductions of neointimal growth, unlike the dose-dependent effect we observed in rats [16], it is possible that the lower fat content of the LFD relative to chow sensitized the vessels to insulin.

Insulin treatment did not decrease plasma glucose levels because insulin-treated mice received 20% glucose in their drinking water. However, as expected, insulin-treated mice showed a decrease in the concentration of TG and FFA, consistent with the plasma TG clearance promotion by insulin and its anti-lipolytic effect [24].

When treated with insulin, mice with EC-specific IR deletion showed greater neointimal growth than IR^f/f^-Cdh5-Cre(−) controls, further demonstrating the vasculoprotective function of insulin in ECs. This beneficial effect of insulin is mainly mediated by eNOS activation according to our previous results showing that the effect is abolished in eNOS knockout mice [11]. Additionally, in our previous studies in rats, we demonstrated that insulin decreases vascular SMC migration and increases re-endothelialization after arterial injury [12], which was abolished by a NOS inhibitor. These results are also in accordance with those found in atherosclerosis models. For instance, a previous study showed that endothelial IR- and apolipoprotein E-knockout mice had increased atherosclerosis [25]. Likewise, we have also reported that insulin treatment in apolipoprotein E knockout mice decreased the atherosclerotic plaque burden, an effect that was abolished by a NOS inhibitor [22].

Similar to EC-specific IR-deficient mice, insulin-treated SMC-specific IR-deficient mice showed a greater neointimal growth than IR^f/f^-SMMHC-Cre(−) controls. These results are consistent with our recent findings in the chow-fed mouse model of restenosis [14], further suggesting that vascular SMC IRs contribute to the anti-stenotic effect of insulin after arterial injury. Although in vitro studies have suggested an adverse effect of insulin on vascular SMC proliferation [26] and migration [8], likely mediated by the MAPK pathway, insulin also inhibits vascular SMC phenotypic switching from dedifferentiated and synthetic to contractile and quiescent via the PI3K pathway [8]. Accordingly, we have reported that insulin treatment enhanced SMC differentiation markers in rat carotid vessels after arterial injury [12]. In our previous study in mice, insulin increased the SMC differentiation marker α-SMA in controls and endothelial-specific IR-deficient mice but failed to increase α-SMA in SMC-IR-deficient mice [14]. Thus, we postulate that insulin action on SMC IRs decreases neointimal formation via an effect on vascular SMC differentiation. Our results supporting the concept of a beneficial effect of insulin on vascular SMCs in vivo are in accordance with recent reports in atherosclerosis models, where the vascular SMC deficiency of IRs accelerated atherosclerosis [27].

We also investigated the effects of insulin on neointimal hyperplasia in EC- or SMC-specific IR-deficient and control mice in HFSD-induced hyperglycemic and insulin-resistant conditions, a model where insulin treatment is clinically relevant. HFSD-fed mice had increased body weight, fed and fasting plasma glucose levels, TG, FFA and total cholesterol levels compared to LFD-fed mice. ITTs indicated HFSD-fed mice were insulin resistant, whereas prior insulin treatment did not affect either insulin resistance or body weight. In accordance with previous reports [28], IGF-1 levels were increased in our HFSD-fed mice. As is the case with insulin, IGF-1 has been found to promote the migration and proliferation of vascular SMCs in vitro [29,30]. However, there are controversial reports about its effects on arterial intimal hyperplasia in vivo [31,32,33]. Notably, fed insulin levels were not increased in HFSD-fed vehicle-treated mice compared to LFD-fed mice, presumably because of HFSD-induced β-cell failure to compensate for insulin resistance, resulting in hyperglycemia. In vehicle-treated HFSD-fed control mice, the neointimal area and I/M ratio were marginally greater than in vehicle-treated LFD controls. This is in accordance with our previous studies in rats where a 60% fat diet [13] or a sucrose diet [16] increased these parameters. Whether HFSD-induced hyperlipidemia, increased IGF-1, or insulin resistance itself was responsible for the greater neointimal growth in HFSD-fed mice compared to LFD-fed mice in the present study remains unclear. Our results in HFSD-fed control mice and EC-specific IR-deficient mice showed that the anti-restenotic effect of insulin treatment was abolished but not reversed, independent of EC-specific IR expression. The findings in control mice are in accordance with our previous results in rats [13], while our data from endothelial-specific IR-deficient mice show that even in the absence of the vasculoprotective effect of insulin in ECs, insulin did not increase neointimal growth. It is possible that the growth-promoting MAPK pathway, which predominates in vascular SMCs and is hypothesized to be unaffected or even potentiated by hyperinsulinemia in conditions of insulin resistance [15], was not activated by insulin in our in vivo conditions. A previous study by our group in insulin-sensitive rats showed decreased vessel MAPK activity after insulin treatment, an effect that was attenuated by NOS inhibition [11]. We and others attributed these findings to an insulin-mediated increase in expression of MAPK phosphatase 1 (MKP-1), which inactivates MAPKs by dephosphorylation [11,34]. Another study reported activation of MAPK in microvessels after insulin stimulation ex vivo but no MAPK activation in the aorta of lean and obese Zucker fa/fa rats in vivo during a hyperinsulinemic-euglycemic clamp, despite greater basal MAPK activation in obese rats [35]. In contrast, it has been previously shown that basal MAPK activation was similar in the aortae of control mice and mice fed with the same HFSD used by us [36]. In the latter study, however, insulin injection in vivo increased MAPK activation (to the same extent in control diet-fed and HFSD-fed mice), though a high bolus insulin dose was used. Since insulin is considered a weak stimulus for vascular cell proliferation compared with other growth factors, such as PDGF [37], it is possible that the mitogenic effect of insulin, mediated by MAPK activation, is not relevant at physiological insulin concentrations in vivo, even under conditions of insulin resistance. Together, these results suggest that the mitogenic MAPK pathway may not directly respond or may even be inhibited by insulin treatment, which could contribute to the abolished but non-reversed anti-restenotic effect of insulin in EC-IR-deficient mice and to the absence of accentuation of the vasculoprotective effect of insulin in SMC-IR-deficient mice under insulin-resistant conditions.

Unexpectedly, but consistent with the results in EC-specific IR-deficient mice, we found that deletion of IRs in SMCs did not have a vasculoprotective effect in either the absence or presence of insulin treatment, further suggesting that insulin’s effect on vascular SMCs does not aggravate neointimal growth. Thus, the fact that subjects with obesity and type 2 diabetes who require insulin have the highest restenosis rates could be attributed to their insulin resistance and relative insulin deficiency rather than their hyperinsulinemia. Our observations conflict with a previous study by another group, in which the authors found a decreased neointimal area in HFD-fed mice with SMC-IR deletion after femoral artery wire injury [17]. However, their reported I/M ratio, which is the gold standard in the assessment of neointimal growth due to vessel size consideration, was not significantly different between SMC-specific IR-deficient and control mice. This study also did not examine the effect of exogenous insulin treatment. It is noteworthy that the same group recently reported a beneficial effect of SMC IR in a model of atherosclerosis, as described above [27].

Our study is not without limitations. First, we did not assess whether insulin signaling was impaired in the vessel in HFSD-fed mice.. An early study in Zucker Fatty rats showed that the IRS/PI3K/Akt pathway was selectively impaired in obese rats compared to lean ones since the protein levels of IRS were decreased in the aorta of obese rats following insulin stimulation. Similarly, the IRS-mediated activity of PI3K and Serine phosphorylation of Akt were reduced in the vessels of obese rats, whereas the Shc/MAPK arm of insulin signaling was unaffected [34]. Ceramide, a lipid metabolite elevated during high-fat feeding, has been demonstrated to diminish insulin signal transduction via PKC and JNK activation [37] and by promoting dephosphorylation of Akt at Ser473 [38]. However, a past study by another group where mice also received a HFSD found that vessel insulin resistance can occur at the level of eNOS [39] without changes in Akt phosphorylation [35,39]. Thus, the site of insulin resistance in diet-induced insulin-resistant conditions should be determined. Further, we previously reported a dose-dependent effect of insulin on neointimal growth in insulin-sensitive rats; however, the dose dependence of the effect of insulin in insulin-resistant conditions remains to be determined. It is possible that a higher insulin dose overcomes vascular insulin resistance and results in a beneficial effect. Alternatively, a higher insulin dose may unmask insulin’s adverse effect on neointimal growth.

Altogether, our results obtained with physiological insulin doses are not in favor of the ‘selective insulin resistance hypothesis’, at least for what concerns the cell-specific effects of insulin. It should be noted that although the PI3K-mediated effects prevail in ECs and the MAPK-mediated effects prevail in SMCs, at least in vitro, insulin engages both pathways in both cells. Therefore, further studies are required to investigate the ‘selective insulin resistance hypothesis’ in the context of the pathway-specific effects of insulin signaling.

## Figures and Tables

**Figure 1 cells-13-01387-f001:**
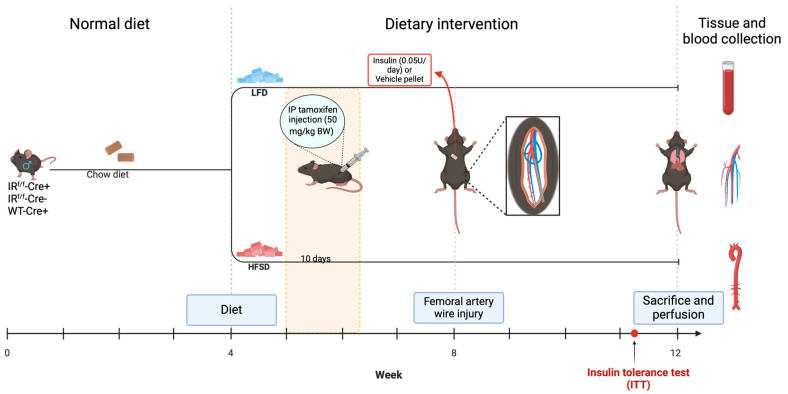
Experimental timeline: Male mice (controls (IR^f/f^-Cre (−), IR^w/w^-Cre (+)), and IR^f/f^-Cre (+)) were fed with either a low-fat diet (LFD) or high fat, high sucrose diet (HFSD) at 4 weeks of age followed by 10 consecutive daily i.p. injections of tamoxifen to induce Cre recombination in the IR^f/f^-Cre(+) mice. Prior to femoral artery wire injury at Week 8, mice were implanted with either an insulin (0.05 U/day) or vehicle pellet. Insulin tolerance tests (ITTs) were conducted five days prior to sacrifice. Body weight and fasted and fed blood glucose measurements were obtained weekly until sacrifice at Week 12, wherein tissue and blood were collected for histological analysis and metabolic parameters, respectively.

**Figure 2 cells-13-01387-f002:**
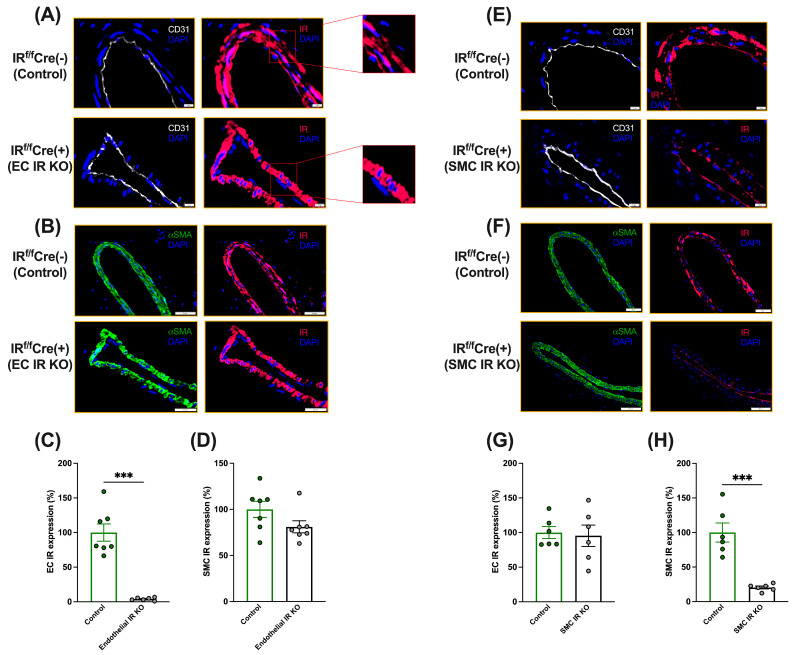
Immunofluorescence analysis of the expression of the insulin receptor (IR) in endothelial cells (ECs) and smooth muscle cells (SMCs) of tamoxifen-treated IR^f/f^-Cdh5-Cre(−) control mice and IR^f/f^-Cdh5-Cre(+)EC-specific IR-deficient mice (**A**–**D**) and IR^f/f^-SMMHC-Cre(−) control mice and IR ^f/f^-SMMHC-Cre(+) SMC-specific IR-deficient mice (**E**–**H**). Uninjured femoral arteries were collected to assess IR expression in ECs and SMCs by immunofluorescence. (**A**,**E**): Representative immunofluorescence images of femoral arteries stained for the IR and the EC marker CD31 (×60), Scale bar: 10 μm. (**B**,**F**): Representative immunofluorescence images for the IR and the SMC marker α-SMA (×20), Scale bar: 50 μm. (**C**,**G**): Expression of IR in endothelial cells (percentage of the average in controls). (**D**,**H**): Expression of IR in smooth muscle cells (percentage of the average in controls). IR^f/f^-Cdh5-Cre (−), *n* = 7; IR^f/f^-Cdh5-Cre (+), *n* = 7, IR^f/f^-SMMHC-Cre (−), *n* = 6; IR^f/f^-Cdh5-Cre (+), *n* = 6. Data are expressed as mean ± SEM. Parametric or non-parametric tests were applied according to the results of normality tests. (**C**,**G**,**H**): Two-tailed unpaired *t*-test: *** *p* < 0.001. (**D**): N.S. Mann–Whitney test.

**Figure 3 cells-13-01387-f003:**
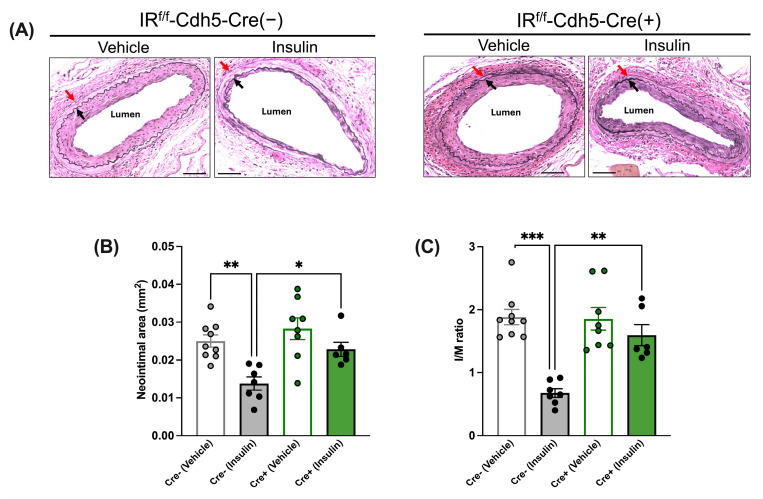
Insulin’s anti-restenotic effect is abolished in tamoxifen-treated low-fat-diet-fed endothelial-cell-specific (EC) insulin receptor (IR)-deficient male mice. (**A**): Representative images of femoral arteries stained with Elastin van Gieson (EVG) (×200). Black arrows indicate internal elastic lamina and red arrows indicate external elastic lamina. (**B**): Neointimal area. (**C**): Intima to media (I/M) ratio. Empty grey and green bars, vehicle; filled grey and green bars, insulin. Scale bar: 50 μm. IR^f/f^-Cdh5-Cre(−) control mice: Vehicle, *n* = 9; Insulin, *n* = 7; IR^f/f^-Cdh5-Cre(+)EC-specific IR-deficient mice: Vehicle, *n* = 8; Insulin, *n* = 6. Data are expressed as mean ± SEM. Parametric or non-parametric tests were applied according to the results of normality tests. One-way ANOVA followed by Tukey’s test: * *p* < 0.05; ** *p* < 0.01; *** *p* < 0.001.

**Figure 4 cells-13-01387-f004:**
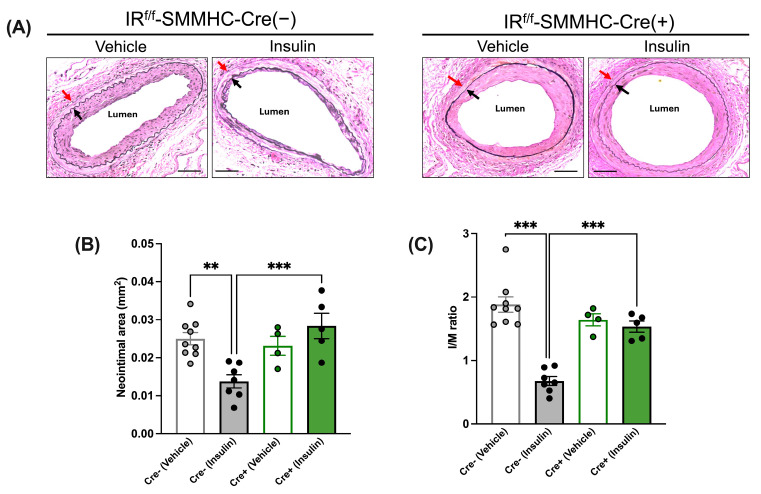
Insulin’s anti-restenotic effect is abolished in tamoxifen-treated low-fat-diet-fed smooth-muscle-cell-specific (SMC) insulin receptor (IR)-deficient male mice. (**A**): Representative images of femoral arteries stained with Elastin van Gieson (EVG) (×200). Black arrows indicate internal elastic lamina and red arrows indicate external elastic lamina. (**B**): Neointimal area. (**C**): Intima to media (I/M) ratio. Empty grey and green bars, vehicle; filled grey and green bars, insulin. Scale bar: 50 μm. IR^f/f^-SMMHC-Cre(−) control mice: Vehicle, *n* = 9; Insulin, *n* = 7; IR^f/f^-SMMHC-Cre(+) SMC-specific IR-deficient mice: Vehicle, *n* = 4; Insulin, *n* = 5. Data are expressed as mean ± SEM. Parametric or non-parametric tests were applied according to the results of normality tests. One-way ANOVA followed by Tukey’s test: ** *p* < 0.01; *** *p* < 0.001.

**Figure 5 cells-13-01387-f005:**
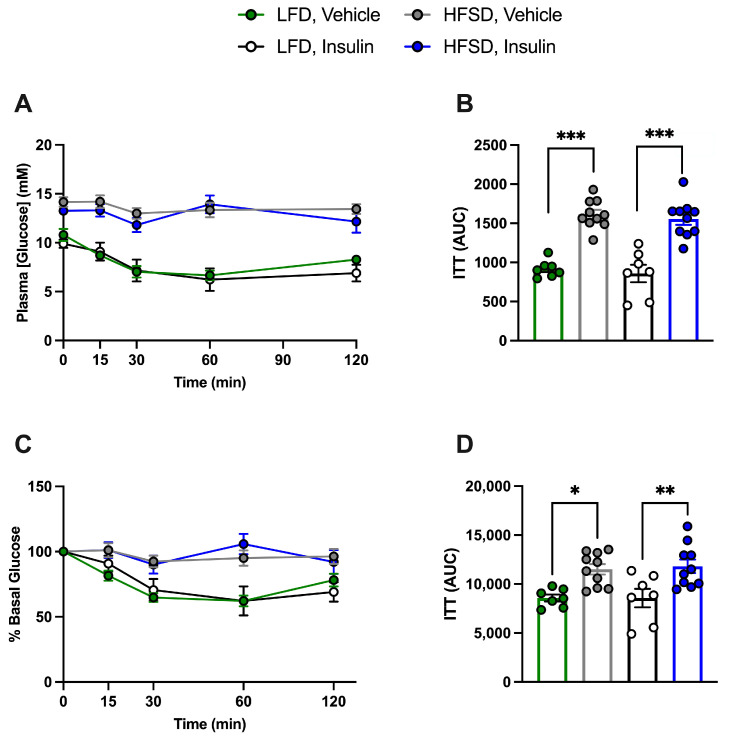
High-fat, high-sucrose-diet-induced insulin resistance in tamoxifen-treated endothelial-cell-specific (EC) insulin receptor (IR)-deficient and control (IR^f/f^-Cre (−)) mice (combined), irrespective of prior insulin treatment. Following 4 h of fasting, an insulin tolerance test (ITT) was conducted on mice, which received an intraperitoneal injection of insulin at 0.5 U/kg. Plasma glucose levels were measured at 0, 15, 30, 60, and 120 min. (**A**): Plasma glucose levels and (**B**): ITT area under the curve (AUC) in (**A**). (**C**): Percentage (%) of basal glucose and (**D**): ITT area under the curve in (**C**). LFD: Low-fat diet; HFSD: high-fat, high-sucrose diet. Data are expressed as mean ± SEM. Parametric or non-parametric tests were applied according to the results of normality tests. (**B**,**D**): One-way ANOVA followed by Tukey’s test: * *p* < 0.05; ** *p* < 0.01; *** *p* < 0.001. *n* > 7 for each group.

**Figure 6 cells-13-01387-f006:**
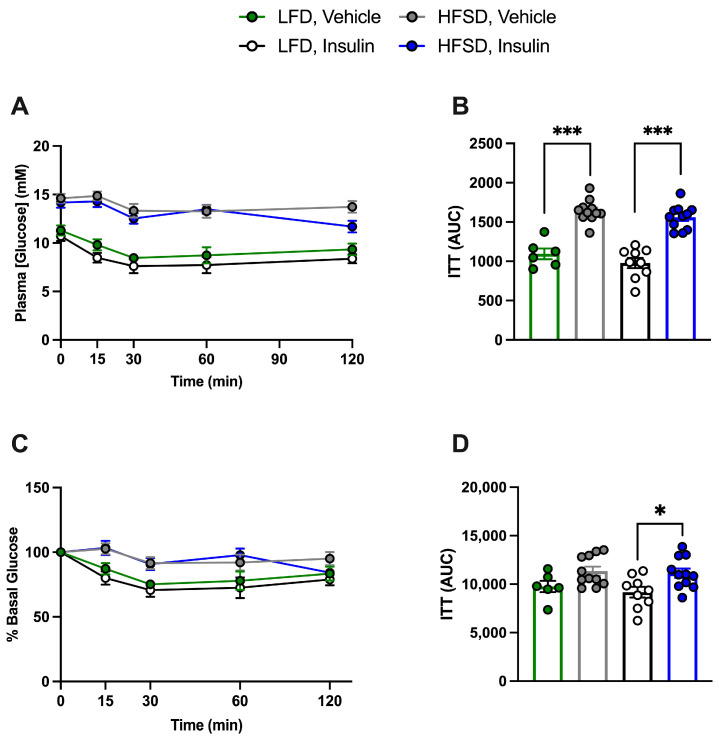
High-fat, high-sucrose-diet-induced insulin resistance in tamoxifen-treated smooth-muscle-cell-specific (SMC) insulin receptor (IR)-deficient and control (IR^f/f^-Cre (−)) mice (combined), irrespective of prior insulin treatment. Following 4 h of fasting, an insulin tolerance test (ITT) was conducted on mice, which received an intraperitoneal injection of insulin at 0.5 U/kg. Plasma glucose levels were measured at 0, 15, 30, 60, and 120 min. (**A**): Plasma glucose levels and (**B**): ITT area under the curve (AUC) in (**A**). (**C**): Percentage (%) of basal glucose and (**D**): ITT area under the curve in (**C**). LFD: Low-fat diet; HFSD: high-fat, high-sucrose diet. Data are expressed as mean ± SEM. Parametric or non-parametric tests were applied according to the results of normality tests. (**B**,**D**): One-way ANOVA followed by Tukey’s test: * *p* < 0.05; *** *p* < 0.001. *n* > 6 for each group.

**Figure 7 cells-13-01387-f007:**
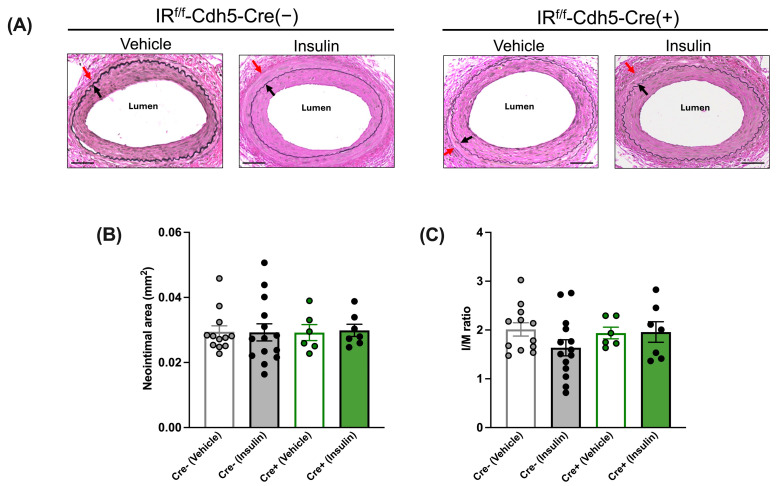
Insulin’s anti-restenotic effect is abolished but not reversed in tamoxifen-treated high-fat, high-sucrose-diet-fed endothelial-cell-specific (EC) insulin receptor (IR)-deficient and control (IR^f/f^-Cre (−)) male mice. (**A**): Representative images of femoral arteries stained with Elastin van Gieson (EVG) (×200). Black arrows indicate internal elastic lamina and red arrows indicate external elastic lamina. (**B**): Neointimal area. (**C**): Intima to media (I/M) ratio. Empty grey and green bars, vehicle; filled grey and green bars, insulin. Scale bar: 50 μm. IR^f/f^-Cdh5-Cre(−) control mice: Vehicle, *n* = 12; Insulin, *n* = 14; IR^f/f^-Cdh5-Cre(+)EC-specific IR-deficient mice: Vehicle, *n* = 6; Insulin, *n* = 7. Data are expressed as mean ± SEM. Parametric or non-parametric tests were applied according to the results of normality tests. (**B**): N.S. Kruskal–Wallis test followed by Dunn’s test. (**C**): N.S. One-way ANOVA followed by Tukey’s test.

**Figure 8 cells-13-01387-f008:**
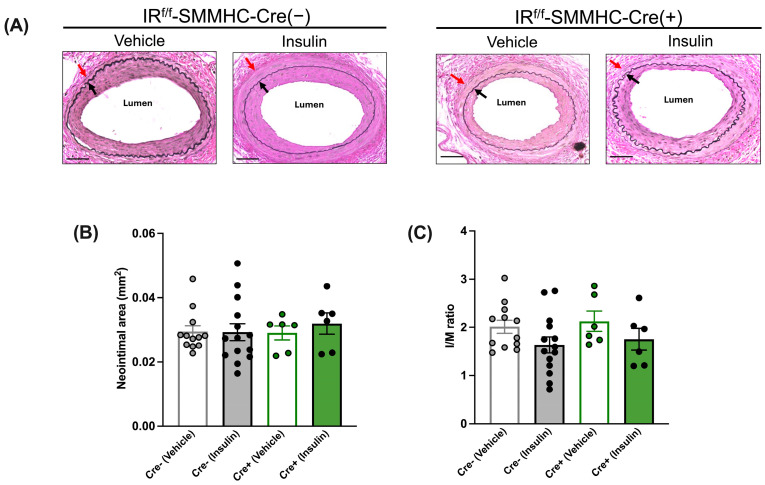
A lack of insulin receptors (IRs) in smooth muscle cells (SMCs) from tamoxifen-treated high-fat, high-sucrose-diet-fed male mice does not decrease neointimal growth in either the absence or presence of insulin treatment. (**A**): Representative images of femoral arteries stained with Elastin van Gieson (EVG) (×200). Black arrows indicate internal elastic lamina and red arrows indicate external elastic lamina. (**B**): Neointimal area. (**C**): Intima to media (I/M) ratio. Empty grey and green bars, vehicle; filled grey and green bars, insulin. Scale bar: 50 μm. IR^f/f^-SMMHC-Cre(−) control mice: Vehicle, *n* = 12; Insulin, *n* = 14; IR^f/f^-SMMHC-Cre(+) SMC-specific IR-deficient mice: Vehicle, *n* = 6; Insulin, *n* = 6. Data are expressed as mean ± SEM. Parametric or non-parametric tests were applied according to the results of normality tests. (**B**): N.S. Kruskal–Wallis test followed by Dunn’s test. (**C**): N.S. One-way ANOVA followed by Tukey’s test.

**Table 1 cells-13-01387-t001:** Metabolic parameters of tamoxifen-treated endothelial-cell-specific IR-deficient (IR^f/f^-Cdh5-Cre (+)) and control (IR^f/f^-Cdh5-Cre (−)) mice. LFD: Low-fat diet; HFSD: High-fat, high-sucrose diet. Data are expressed as mean ± SEM. Parametric or non-parametric tests were applied according to the results of normality tests. Three-way ANOVA followed by Tukey’s test: significant diet main effect was found for body weight, fasting plasma glucose, fed plasma glucose, IGF-1, triglycerides, free fatty acids, and total cholesterol (* *p* < 0.05); significant treatment main effect was found for insulin, triglycerides, and free fatty acids (^$^ *p* < 0.001). *n* > 6 for each group.

	LFD, Vehicle		LFD, Insulin		HFSD, Vehicle		HFSD, Insulin	
	IR^f/f^ Cre (−)	IR^f/f^ Cre (+)	IR^f/f^ Cre (−)	IR^f/f^ Cre (+)	IR^f/f^ Cre (−)	IR^f/f^ Cre (+)	IR^f/f^ Cre (−)	IR^f/f^ Cre (+)
Body Weight (g)	25.7 ± 0.7	27.4 ± 0.7	26.1 ± 0.5	25.5 ± 0.6	31.4 ± 0.7 *	33 ± 1.2 *	30.2 ± 0.6 *	29.3 ± 0.9 *
Fasted Plasma Glucose (mM)	11.1 ± 0.6	10.0 ± 0.6	9.9 ± 0.5	9.9 ± 0.6	14.6 ± 0.6 *	14 ± 0.8 *	13.8 ± 0.3 *	13 ± 0.4 *
Fed Plasma Glucose (mM)	10.5 ± 0.3	10.1 ± 0.2	10.4 ± 0.2	9.5 ± 0.3	12.9 ± 0.2 *	12.3 ± 0.4 *	12.5 ± 0.4 *	12.7 ± 0.7 *
Fed Insulin (pM)	200 ± 85	200 ± 58	540 ± 120 ^$^	469 ± 88 ^$^	131± 23	88 ± 12	474± 110 ^$^	507 ± 148^$^
Fed IGF-1 (pg/mL)	506 ± 81	404 ± 90	472 ± 45	402 ± 36	776 ± 64 *	623 ± 79 *	604 ± 60 *	570 ± 46 *
Fed VEGF (pg/mL)	21.7 ± 1.5	24.9 ± 1.43	23.4 ± 1.44	25.7 ± 1.86	25.5 ± 1.39	23.2 ± 1.42	27.7 ± 2.10	26.7 ± 1.70
Fed Triglycerides (mM)	1.4 ± 0.1	1.6 ± 0.1	1.1 ± 0.1 ^$^	0.8 ± 0.2 ^$^	1.7 ± 0.1 *	1.5 ± 0.2 *	1.3 ± 0.1 *^$^	1.3 ± 0.2 *^$^
Fed Free Fatty Acids (mM)	0.63 ± 0.04	0.58 ± 0.04	0.52 ± 0.03 ^$^	0.49 ± 0.04 ^$^	0.75 ± 0.05 *	0.66 ± 0.04 *	0.59 ± 0.02 *^$^	0.58 ± 0.02 *^$^
Total cholesterol (mM)	1.4 ± 0.18	1.3 ± 0.17	1.4 ± 0.31	1.4 ± 0.43	2.7 ± 0.31 *	2.3 ± 0.38 *	3.1 ± 1.0 *	2.3 ± 0.44 *

**Table 2 cells-13-01387-t002:** Metabolic parameters of tamoxifen-treated smooth-muscle-cell-specific IR-deficient (IR^f/f^-SMMHC-Cre (+)) and control (IR^f/f^-SMMHC-Cre (−)) mice. LFD: Low-fat diet; HFSD: High-fat, high-sucrose diet. Data are expressed as mean ± SEM. Parametric or non-parametric tests were applied according to the results of normality tests. Three-way ANOVA followed by Tukey’s test: significant diet main effect was found for body weight, fasting plasma glucose, fed plasma glucose, IGF-1, triglycerides, free fatty acids, and total cholesterol (* *p* < 0.05); significant treatment main effect was found for insulin, triglycerides, and free fatty acids (^$^ *p* < 0.05). *n* > 6 for each group.

	LFD, Vehicle		LFD, Insulin		HFSD, Vehicle		HFSD, Insulin	
	IR^f/f^Cre (−)	IR^f/f^ Cre (+)	IR^f/f^ Cre (−)	IR^f/f^ Cre (+)	IR^f/f^ Cre (−)	IR^f/f^ Cre (+)	IR^f/f^ Cre (−)	IR^f/f^ Cre (+)
Body Weight (g)	25.7 ± 0.7	26 ± 0.5	26.1 ± 0.5	25.8 ± 0.5	31.4 ± 0.7 *	29.4 ± 0.4 *	30.2 ± 0.6 *	31 ± 0.4 *
Fasted Plasma Glucose (mM)	11.1 ± 0.6	11.2 ± 0.3	9.9 ± 0.5	10.3 ± 0.7	14.6 ± 0.6 *	14.6 ± 0.7 *	13.8 ± 0.3 *	14.5 ± 0.9 *
Fed Plasma Glucose (mM)	10.5 ± 0.3	10.4 ± 0.3	10.4 ± 0.2	9.9 ± 0.6	12.8 ± 0.2 *	12.5 ± 0.4 *	12.8 ± 0.4 *	12.2 ± 0.8 *
Fed Insulin (pM)	200 ± 85	111 ± 26	540 ± 120 ^$^	439 ± 103 ^$^	131 ± 23	87 ± 12	474 ± 110 ^$^	344 ± 55 ^$^
Fed IGF-1 (pg/mL)	506 ± 81	562 ± 33	472 ± 45	376 ± 53	776 ± 64 *	650 ± 14 *	604 ± 60 *	560 ± 44 *
Fed VEGF (pg/mL)	21.7 ± 1.5	25.5 ± 1.39	23.4 ± 1.44	27.7 ± 2.10	25.5 ± 1.39	24.2 ± 1.44	27.7 ± 2.10	27.2 ± 1.09
Fed Triglycerides (mM)	1.4 ± 0.1	1.5 ± 0.2	1.1 ± 0.1 ^$^	0.9 ± 0.1 ^$^	1.7 ± 0.1 *	1.5 ± 0.1 *	1.3 ± 0.1 *^$^	1.5 ± 0.1 *^$^
Fed Free Fatty Acids (mM)	0.63 ± 0.04	0.54 ± 0.04	0.52 ± 0.03 ^$^	0.40 ± 0.04 ^$^	0.75 ± 0.05 *	0.69 ± 0.02 *	0.59 ± 0.02 *^$^	0.63 ± 0.02 *^$^
Total cholesterol (mM)	1.4 ± 0.18	1.7 ± 0.21	1.4 ± 0.31	1.1 ± 0.27	2.7 ± 0.31 *	2.0 ± 0.16 *	3.1 ± 1.0 *	2.1 ± 0.19 *

## Data Availability

The original contributions presented in the figures are included in the article/Appendix A, further inquiries can be directed to the corresponding author.

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
