# Peer review of "Cell-Specific Effects of Insulin in a Murine Model of Restenosis Under Insulin-Sensitive and Insulin-Resistant Conditions"

_cells, 2024, doi:10.3390/cells13161387_

Round 1

Reviewer 1 Report

Comments and Suggestions for Authors

Restenosis refers to the reduction in vessel lumen diameter after percutaneous recanalization.  Despite advances in stent technology, restenosis remains the most common cause of target lesion failure after percutaneous coronary intervention. Restenosis usually occurs within 3 to 6 months after PCI and is defined as vessel lumen narrowing that again exceeds 30%. This study is quite interesting, but I have some minor concerns:

1)         Line 34  defineI/M ratiowhen they first appeared in the manuscript. please check the whole manuscript

2)         2.1. Animal Model and Animal Experiments

The author should clearly explain how many mice were used in this study and how they were grouped. In Figure 1 Mice (controls and IRf/f-Cre+) were fed with either LFD or HFSD, controls mean what? The author should clarify the entire experimental grouping and operation process for non-professionals to understand. 

3)         Figure1 C-D, G-H, why some of the percentage of IR expression(%) over 100%? How they measured? please clarify

4)         Line 139 1 hour should be "1 h",

5)         Figure legends please define the abbreviations for example, Figure 1 define the abbreviation of LFD and HFSD, EVG in Figure S 2-5

6)         Line 290 ".." keep one "."

7)         Line 352 and 383 " in vitro " italic, please check the whole manuscript

8)         Line 378-379, there is no evidence of PI3K/Akt/eNOS pathway measured in ECs in this study. Please consider whether the sentence is appropriate here.

Author Response

Reviewer 1

Comments and Suggestions for Authors

Restenosis refers to the reduction in vessel lumen diameter after percutaneous recanalization. Despite advances in stent technology, restenosis remains the most common cause of target lesion failure after percutaneous coronary intervention. Restenosis usually occurs within 3 to 6 months after PCI and is defined as vessel lumen narrowing that again exceeds 30%. This study is quite interesting, but I have some minor concerns:

1-Line 34 define “I/M ratio”when they first appeared in the manuscript. please check the whole manuscript

Thank you for pointing this out. Intima to media (I/M) ratio has now been defined the first time it appears in the manuscript on lines 34-35.

2-Animal Model and Animal Experiments 

The author should clearly explain how many mice were used in this study and how they were grouped. In Figure 1 Mice (controls and IRf/f-Cre+) were fed with either LFD or HFSD, controls mean what? The author should clarify the entire experimental grouping and operation process for non-professionals to understand. 

Thank you for the comment. In lines 53-59 of the methodology section, we have explained the groups of mice that we used for the study, IRf/f-Cdh5-Cre(+) and IRf/f-SMMHC-Cre(+) mice and control mice. Our control mice were IRf/f-Cre(-) and Cdh5- or SMMHC- IRw/w-Cre(+). When treated with tamoxifen, IRf/f-Cdh5-Cre(+) and IRf/f-SMMHC-Cre(+) mice become insulin receptor (IR) deficient in ECs and SMCs respectively (line 79). We have further defined our groups in each Figure legend where we have also reported the number of animals. In addition, we have now consistently used the term ‘Control’ for mice and the term ‘Vehicle’ for treatment.

3-Figure1 C-D, G-H, why some of the percentage of IR expression(%) over 100%? How they measured? please clarify

Thanks for the comment. We averaged the values of the control (IRf/f-Cre(-)) animals, and then divided the value of each animal in the control group by the average as now specified. In this way we can show the variability of the data where the average is 100%. The values of the knockout (IRf/f-Cre(+)) mice were then normalized to the average value of the controls.

4-Line 139 “1 hour” should be "1 h", 

Thanks for pointing this out. We have now corrected this.

5-Figure legends please define the abbreviations for example, Figure 1 define the abbreviation of LFD and HFSD, EVG in Figure S 2-5

Thanks for pointing this out. We have now defined the abbreviations in all the figures.

6-Line 293 ".." keep one "."

Thanks for pointing this out. We have now corrected this.

7-Line 352 and 383 " in vitro " italic, please check the whole manuscript.

Thank you for the comment. All the “in vivo” and “in vitro” have been italicized.

8-Line 378-379, there is no evidence of PI3K/Akt/eNOS pathway measured in ECs in this study. Please consider whether the sentence is appropriate here.

We agree with this point. We have now modified this to just “the vasculoprotective effect of insulin” in ECs.

Reviewer 2 Report

Comments and Suggestions for Authors

In the present study, Dr. Marel Gonzalez-Medina and colleagues aimed to study the cell-specific effects of insulin on neointimal growth in an in vivo murine model of restenosis under insulin-sensitive and insulin-resistant conditions by using the CreERT2-loxP-mediated recombination system, which can selectively and conditionally induce IR deletion in ECs or SMCs without affecting tissue development. The authors reported that insulin decreased neointimal growth in insulin-sensitive conditions only in controls. In contrast, under insulin-resistant conditions, insulin had no effect in either controls, EC-specific, or SMC-specific IR-deficient mice suggesting that IRs of EC and SMC are required for the anti-restenosis impact of insulin in insulin-sensitive conditions and that in insulin resistance, insulin has no adverse effect on vascular SMC in vivo.

Overall, this is a clear, concise, and well-written manuscript. The methods are generally appropriate, although clarification of a few details and a rationale for using this particular method of measuring eating disorder symptoms should be provided. Overall, the results are clear and compelling.

Specific comments:

1/ It's not clear if the authors have used male or female mice. Authors should explain this aspect taking into consideration the sex difference in insulin resistance in HFD-induced prediabetes/diabetes in mouse models (Female seems to be more protected against HFD-induced diabetes).

2/ Did the authors attempt to generate the double knockout mice IRs of both EC and SMCs?

3/ Did the authors perform the index insulin resistance test HOMA (Homeostatic Model Assessment) to assess insulin resistance levels in these mice?

4/ Data for circulating makers for cell migration and growth might be relevant (eg. VEGF, IGF1, etc.)

5/ If available, the addition of lipid profile data might be relevant since studies have supported lipid changes in the neointima during restenosis.

Author Response

Reviewer 2

Comments and Suggestions for Authors

In the present study, Dr. Marel Gonzalez-Medina and colleagues aimed to study the cell-specific effects of insulin on neointimal growth in an in vivo murine model of restenosis under insulin-sensitive and insulin-resistant conditions by using the CreERT2-loxP-mediated recombination system, which can selectively and conditionally induce IR deletion in ECs or SMCs without affecting tissue development. The authors reported that insulin decreased neointimal growth in insulin-sensitive conditions only in controls. In contrast, under insulin-resistant conditions, insulin had no effect in either controls, EC-specific, or SMC-specific IR-deficient mice suggesting that IRs of EC and SMC are required for the anti-restenosis impact of insulin in insulin-sensitive conditions and that in insulin resistance, insulin has no adverse effect on vascular SMC in vivo.

Overall, this is a clear, concise, and well-written manuscript. The methods are generally appropriate, although clarification of a few details and a rationale for using this particular method of measuring eating disorder symptoms should be provided. Overall, the results are clear and compelling.

Specific comments:

1- It's not clear if the authors have used male or female mice. Authors should explain this aspect taking into consideration the sex difference in insulin resistance in HFD-induced prediabetes/diabetes in mouse models (Female seems to be more protected against HFD-induced diabetes).

Thank you for this comment. We used male mice for the study because the SMMHC-Cre transgene used is located on the Y chromosome (Groneberg, D et al, Circulation 121:401-409, 2010). We have now clarified this in the manuscript on Line 59. Since there is now a newly developed transgene located on chromosome 2, it would be interesting to perform this study in females that are less susceptible than males to HFD metabolic disturbances, as stated by the reviewer, and also display less extensive neointimal growth after arterial injury (PMID: 35715609; PMID: 32324497).

2- Did the authors attempt to generate the double knockout mice IRs of both EC and SMCs?

This is a very important comment. Thank you for bringing it up. Since these are time-consuming studies, we have not generated the double knockout mice. However, by doing so in our future studies, we might obtain new information about the cell-specific effects of insulin in models of restenosis. Since the effect of insulin in LFD was abolished by deleting the IR from either ECs or SMCs, we might speculate that the results in the presence of insulin would not be different from the single knockouts, however it is possible that double knockouts would show greater neointimal growth even when vehicle treated.

3- Did the authors perform the index insulin resistance test HOMA (Homeostatic Model Assessment) to assess insulin resistance levels in these mice? 

Thank you for the comment. We could not perform HOMA as we did not measure fasting insulin because we only had fed plasma. However, we assessed insulin sensitivity by insulin tolerance test, and as shown in the results (Figures 5 and 6), mice fed a HFSD were insulin resistant compared to those fed LFD regardless of treatment (insulin vs. vehicle) or genotype.

4- Data for circulating markers for cell migration and growth might be relevant (eg. VEGF, IGF1, etc.) 

As suggested by the reviewer, we performed a VEGF and IGF-1 ELISA and included the results in Tables 1 and 2. We found no changes in circulating VEGF levels in our mice regardless of genotype, treatment, and diet. We had previously found that insulin treatment resulted in no significant change in VEGF mRNA in injured carotids (Guo et al, PMID: 25974101). In terms of IGF-1, we found a significant diet main effect following a three-way ANOVA, with no differences regarding genotype or treatment (vehicle vs. insulin). As briefly mentioned in the paper, although IGF-1 stimulates vascular smooth muscle cell proliferation and migration in vitro, the in vivo effect of IGF-1 in models of restenosis is controversial, as the literature has implicated IGF-1 signaling in both promoting (PMID:11459808; PMID:15541402) and inhibiting (PMID:19740101) arterial intimal hyperplasia.

5- If available, the addition of lipid profile data might be relevant since studies have supported lipid changes in the neointima during restenosis.

Thank you for the suggestion. In addition to our initial analysis of free fatty acids and triglycerides, we have added total cholesterol levels to Tables 1 and 2, where we observe a significant main diet effect after doing a three-way ANOVA. This is in accordance with our free fatty acids and triglycerides results, where HFSD-fed mice had elevated triglyceride and free fatty acid levels compared to LFD-fed mice. We could not perform lipoprotein fractionation as we only had frozen plasma.

Round 2

Reviewer 2 Report

Comments and Suggestions for Authors

The authors have successfully addressed all my previous comments/suggestions.

No further comments. Thank you!